# MiRNAs and Microbiota in Non-Small Cell Lung Cancer (NSCLC): Implications in Pathogenesis and Potential Role in Predicting Response to ICI Treatment

**DOI:** 10.3390/ijms25126685

**Published:** 2024-06-18

**Authors:** Francesco Nucera, Paolo Ruggeri, Calogera Claudia Spagnolo, Mariacarmela Santarpia, Antonio Ieni, Francesco Monaco, Giovanni Tuccari, Giovanni Pioggia, Sebastiano Gangemi

**Affiliations:** 1Pneumologia, Dipartimento di Scienze Biomediche, Odontoiatriche e delle Immagini Morfologiche e Funzionali (BIOMORF), Università degli Studi di Messina, 98166 Messina, Italy; francesco.nucera@unime.it; 2Medical Oncology Unit, Department of Human Pathology “G. Barresi”, University of Messina, 98122 Messina, Italy; spagnoloclaudia92@gmail.com (C.C.S.); mariacarmela.santarpia@unime.it (M.S.); 3Department of Human Pathology in Adult and Developmental Age “Gaetano Barresi”, Section of Anatomic Pathology, University of Messina, 98100 Messina, Italy; aieni@unime.it (A.I.); giovanni.tuccari@unime.it (G.T.); 4Chirurgia Toracica, Dipartimento di Scienze Biomediche, Odontoiatriche e delle Immagini Morfologiche e Funzionali (BIOMORF), Università degli Studi di Messina, 98166 Messina, Italy; francesco.monaco@unime.it; 5Institute for Biomedical Research and Innovation (IRIB), National Research Council of Italy (CNR), 98164 Messina, Italy; giovanni.pioggia@cnr.it; 6Operative Unit of Allergy and Clinical Immunology, Department of Clinical and Experimental Medicine, University of Messina, Via Consolare Valeria 1, 98125 Messina, Italy; sebastiano.gangemi@unime.it

**Keywords:** NSCLC, lung/gut microbiota, miRNAs, ICI treatment, biomarkers

## Abstract

Lung cancer (LC) is one of the most prevalent cancers in both men and women and today is still characterized by high mortality and lethality. Several biomarkers have been identified for evaluating the prognosis of non-small cell lung cancer (NSCLC) patients and selecting the most effective therapeutic strategy for these patients. The introduction of innovative targeted therapies and immunotherapy with immune checkpoint inhibitors (ICIs) for the treatment of NSCLC both in advanced stages and, more recently, also in early stages, has revolutionized and significantly improved the therapeutic scenario for these patients. Promising evidence has also been shown by analyzing both micro-RNAs (miRNAs) and the lung/gut microbiota. MiRNAs belong to the large family of non-coding RNAs and play a role in the modulation of several key mechanisms in cells such as proliferation, differentiation, inflammation, and apoptosis. On the other hand, the microbiota (a group of several microorganisms found in human orgasms such as the gut and lungs and mainly composed by bacteria) plays a key role in the modulation of inflammation and, in particular, in the immune response. Some data have shown that the microbiota and the related microbiome can modulate miRNAs expression and vice versa by regulating several intracellular signaling pathways that are known to play a role in the pathogenesis of lung cancer. This evidence suggests that this axis is key to predicting the prognosis and effectiveness of ICIs in NSCLC treatment and could represent a new target in the treatment of NSCLC. In this review, we highlight the most recent evidence and data regarding the role of both miRNAs and the lung/gut microbiome in the prediction of prognosis and response to ICI treatment, focusing on the link between miRNAs and the microbiome. A new potential interaction based on the underlying modulated intracellular signaling pathways is also shown.

## 1. Introduction

Lung cancer (LC) is one of the most frequent cancers diagnosed in both men and women and is still correlated with high mortality and lethality. Non-small cell lung cancer (NSCLC) is the most frequently diagnosed type of LC, representing ~85% of all cases, compared with small-cell lung cancer (SCLC, 15% of all cases) [1]. The pathogenesis of NSCLC is very complex, involving the impaired activation of several intracellular signaling pathways, such as phosphatidylinositol 3-kinase (PI3K) and mitogen-activated protein kinases (MAPK), epidermal growth factor receptor (EGFR), phosphatidylinositol 3-kinase catalytic alpha (PIK3CA), and phosphatase and tensin homolog (PTEN) pathways. The different activation/impairment of these molecular pathways and also the presence of molecular and genetic differences can mediate the heterogeneous characteristics of this disease [1].

Recent advances in the field of molecular biology have led to the identification of several biomarkers aimed at selecting the most appropriate therapeutic strategy for NSCLC patients. In fact, therapeutic options for these diseases have recently notably been improved by the introduction of innovative treatments, including targeted therapies and immunotherapy, both in the advanced stages and, more recently, also in the early stages. Currently, targeted therapy represents the standard of care for NSCLC patients with driver mutations, while immune checkpoint inhibitors (ICIs) as single agents or in combination with chemotherapy represent an important therapeutic approach for patients with non-oncogene addicted NSCLC.

However, despite the positive results obtained with ICIs, only a proportion of NSCLC patients show clinical benefits from these agents, and several patients can experience disease progression [2,3]. Many studies have focused on the identification and role of potential predictive biomarkers that could be used to carefully select patients for ICI treatment. Currently, there are a few approved predictive biomarkers such as programmed death-ligand 1 (PD-L1) expression, tumor mutational burden (TMB), and microsatellite instability (MSI) [2]. Both micro-RNAs (miRNAs) and the lung microbiome seem to be involved in the modulation of pathways regulating the recruitment and activation of several immune cells. They could thus represent new and useful prognostic and predictive factors for drug effectiveness.

Several studies have evaluated miRNAs activity in NSCLC, focusing on its correlation with patient outcomes and drug effectiveness [4,5,6,7,8]. On the other hand, there is some evidence highlighting the key role of the microbiome in the pathogenesis of several chronic inflammatory diseases and LC. This seems to be correlated with the capability of microbiota and the related microbiome to modulate the inflammatory status through the interaction with the host immune cells [9]. A change in the composition of the microbiota, so-called dysbiosis, can play an important role in the evolution and maintenance of several respiratory diseases, including LC, as this can impair the host immune response, thus decreasing the capacity of the immune cells to recognize and kill tumor cells [9,10]. In this review, we present the latest evidence on the role of microbiota and miRNAs in the pathogenesis and prognosis of NSCLC. We also focus on their possible interactions and their capacity to modulate the host immune response in order to understand their possible role in predicting the benefits of ICI treatment for NSCLC patients.

## 2. Physiology, Activity, and Classification of MiRNAs

MiRNAs are a large family of non-coding RNAs. These are divided according to the molecular size and structure into long non-coding RNA (lncRNAs), characterized by >200 nucleotides, and small non-coding RNAs (sncRNAs), characterized by <200 nucleotides including miRNAs, which are usually composed by 21–23 nucleotides [1]. miRNAs can be located in tissue (tissue-specific miRNAs) and are directly obtained from any kind of tissue (including lung tissue) through RNA-sequencing. Alternatively, these may freely circulate in different body fluids (including serum and are referred to as circulating miRNAs) or be stored in extracellular vesicles (EVs), such as exosomes, which protect the miRNAs from degradation [11].

miRNAs are the most prevalent class of sncRNAs. They play a role in post-transcriptional gene modification through their complementary binding to messenger RNA (mRNA), modulating the translation process [1,12].

The mechanisms mediated by miRNAs include:-Elongation suppression (also termed mRNAs cleavage or repressed mRNAs);-Inhibition of translation (also termed Cap and 60 S joining inhibition);-Ribosome drop-off (also termed premature termination);-Co-translational protein degradation [13].

The production and expression of several miRNAs in cancer cells differ widely from the miRNAs expressed by normal cells, thus suggesting that these molecules may play a role in several impaired processes in tumors such as tumor growth, angiogenesis, and immune evasion [14].

## 3. Regulation of miRNAs in Key Signaling Pathways Involved in NSCLC

All possible alterations in the sequences of any miRNA may play a key role in the pathogenesis of different neoplasms, including LC. miRNAs modulate several intracellular signaling pathways and induce many effects. These molecules can be classified according to their effects on tumor-suppressive miRNAs and oncogenic miRNAs (oncomirs) [12,15].

### 3.1. MiRNAs Are Involved in the Regulation of the Tumor-Suppressive Signaling Pathway

The retinoblastoma (RB) pathway is key to regulating cell cycle and cell death through the modulation of several intracellular signaling pathways such as cyclin (CCD), cyclin-dependent kinase (CDK), and E2F-family transcription factors [16]. The RB signaling pathway can be inhibited in cancers through three processes: decreased RB protein function, increased CDK expression, and decreased expression of CDK inhibitors [17]. Inhibition of these oncogenic proteins such as CCN, CDK, and E2F1/3 in tumor cells thus represents a potential novel approach for the treatment of NSCLC [16]. For example, both miRNA-671-3p and miRNA-646 can target CCND2 mRNA, decreasing CCND2 levels, thus inhibiting the proliferation and invasion capacity of NSCLC cells [18,19]. In addition, decreased levels of both miRNA-34b-3p and miRNA-340 have been found to characterize NSCLC cells and correlate positively with metastasis and tumor size. In addition, the application of miRNAs to tumor cells decreased proliferation, through cell cycle arrest, and enhanced apoptosis [20,21].

### 3.2. Involvement of miRNAs in the Regulation of Oncogenic Signaling Pathways

The PI3K pathway also plays a key role in the pathogenesis of neoplasms as it can modulate several biological mechanisms of cancer cells such as tumor proliferation, growth, and survival [17]. The dysregulation of this signaling pathway can induce oncogenic effects correlated with the development of several human cancer types, including lung cancer (through enhanced cell proliferation), metastasis, angiogenesis, epithelial-to-mesenchymal transition (EMT), chemoresistance, and apoptosis [17]. There are several miRNAs that can modulate the PI3K signaling pathway, thus regulating lung cancer development. One study showed that in vitro use of miRNA-126 that can target PI3K led to decreased AKT phosphorylation and inhibited A549 cell proliferation, metastasis, and invasion [22].

Another in vivo study confirmed these data, showing that the use of exosomes in a mouse model led to decreased NSCLC growth and metastasis risk [23]. In addition, the enhanced expression of both miRNA-133b and miRNA-145 mediated the decreased activation of the PI3K signaling pathways through the reduced expression of the epidermal growth factor receptor (EGFR) in A549 cells, which correlated with decreased cell proliferation and increased cell apoptosis [24,25]. The mechanistic target of the rapamycin (mTOR) protein also played a key role in several cell processes such as proliferation, metastasis, and, in particular, radio sensitivity [26]. Several in vitro and in vivo studies have shown that miRNAs such as miRNA-99a and miRNA-101-3p can decrease mTOR expression, increasing the radio sensitivity of NSCLC cells [27,28].

The RTK/RAS/MAPK signaling pathway is another important intracellular oncogenic signaling pathway. It can modulate several cellular processes such as cell proliferation, migration, and apoptosis through the activation of other correlated signaling pathways such as the Janus kinase/signal transducer and activator of transcription (JAK/STAT), mitogen-activated protein kinase/extracellular signal-regulated kinase (MAPK/ERK), and rat sarcoma virus/rapidly accelerated fibrosarcoma pathways (RAS/RAF) [17].

Another study highlighted that, through the in vitro use of miRNA-148a–3p, inhibition of the mitogen-activated protein kinase kinase kinase 9 (MAP3K9) decreased tumor development, cytoskeleton remodeling, and the EMT process [29].

The Wnt signaling pathway has a key role in the modulation of several intracellular processes, including cell motility, cell polarity, cell proliferation, and differentiation [17]. Targeting this signaling pathway could thus play a potential role in the treatment of NSCLC. In vitro studies have highlighted the role of both miRNA-1253 and miRNA-577 as tumor suppressors through the inhibition of WNT5A and WNT2A, respectively, which block the proliferation of lung cancer cell, migration, and the EMT process [30,31]. Other in vitro studies have shown that the use of miRNA-512-5p and miR-100 decreases the expression of β-catenin [32,33].

### 3.3. MiRNAs Have Dual Roles in Tumorigenic Signaling Pathways

The transforming growth factor-β (TGF-β) signaling pathway regulates several cellular processes such as cell differentiation, cell proliferation, and cell-specific and tissue-specific in motility embryonic and adult tissues through modulation of gene-specific expression, RNA processing, and mRNA translation [17]. Depending on the lung cancer stage, the TGF-β signaling pathway can mediate opposite effects in neoplasm cells. In early stages, TGF-β acts as a tumor suppressor decreasing cell growth and enhancing apoptosis, while, in late stages, TGF-β mediates metastasis development and enhances the EMT process [17].

Data show how the in vitro use of miRNA-454-3p and miRNA-17-5p decreases tumor cell proliferation and enhances apoptosis through the inhibition of the TGF-β2 expression [34,35]. Other data have also shown that the in vitro use of miRNA-769-5p induces decreased expression of TGF-β receptors, inhibiting the proliferation and migration of tumor cells [36].

## 4. Role of miRNAs as Prognostic Factors in Patients with NSCLC

Several studies have focused on the role of circulating and tissue miRNAs, which represent new potential biomarkers to predict the prognosis of patients with NSCLC [37,38].

### 4.1. Circulating miRNAs

Serum miRNA-30 levels were found to be lower in NSCLC patients compared with healthy donors. In addition, lower levels of serum miRNA-30 in NSCLC patients were correlated to a shorter median overall survival rate (OS, 23.0 months), compared with NSCLC patients with increased miRNA-30 serum levels (OS, 36.0 months). Moreover, serum miRNA-30 levels were negatively correlated with the disease stage, tumor size, and the presence of both tumor node and lymph node metastasis [39]. NSCLC patients (both with early and advanced stages) characterized by increased serum exosomal miRNA-1246 levels have been shown to have a lower OS and disease-free survival (DFS) rate compared with NSCLC patients with lower serum exosomal miRNA-1246 levels [40]. Serum miRNA-762 levels have been shown to correlate positively with a worse prognosis in NSCLC patients. Upregulation of miRNA-762 was found in NSCLC patients characterized by advanced clinical stages (III and IV), lymph node metastasis, and low tumor differentiation, as serum miRNA-762 levels also negatively correlated with histological grade [41]. Serum expression was found to decrease compared with healthy donors. Serum miRNA-519d levels were found to correlate negatively with lymph node metastasis, advanced stage, and significantly decreased OS [42].

Another study showed that both serum miRNA-942 and miRNA-601 levels were higher in NSCLC patients (both with early and advanced stages) compared with the control. Moreover, serum miRNAs levels e (miR-942 and miR-601) were positively correlated with poor survival of NSCLC patients [43]. A study of an Indian population of NSCLC patients evaluated the serum of both miRNA-375 and miRNA-10b-5p. Serum levels of both miRNAs were decreased in NSCLC patients (both with early and advanced stages) compared with healthy controls. Serum miRNA-375 levels also correlated negatively with lymph node metastasis (*p* = 0.0224) and with pleural effusion (*p* = 0.0148). Conversely, miRNA-10b-5p serum levels were negatively correlated only with the pleural effusion (*p* = 0.0037) [44]. Increased serum miRNA-518b levels correlated positively with tumor size (*p* = 0.042), advanced TNM stage (*p* = 0.006), and lymph node metastasis (*p* = 0.039). In addition, serum miRNA-518b levels correlated positively with shorter survival of patients (*p* = 0.009) [45]. Decreased serum miRNA-185 levels correlated with poor survival rate and adverse clinicopathological parameters in NSCLC patients (both in early and advanced stages), which represents an independent prognostic factor in NSCLC patients [46]. Another study showed that serum levels of miRNA-1228-3p and miRNA-181a-5p were significantly associated with OS in NSCLC levels with a positive and negative correlation, respectively [47]. Increased serum miRNA-3195 levels in NSCLC patients were characterized by longer OS (*p* = 0.0298) compared with NSCLC patients with decreased miRNA-3195 levels [44].

Increased miR-21-5p and decreased miRNA-126-3p serum levels were found in NSCLC patients compared with healthy controls. Enhanced OS was also found in NSCLC patients, characterized by both lower miRNA-21-5p and higher miRNA-126-3p serum levels [48].

Another study showed that serum miRNA-382 levels were lower than healthy controls; however, after surgical resection, these patients showed an increase in terms of serum miRNA-382 expression [49]. Serum levels were found to be upregulated in NSCLC compared with healthy controls. In addition, serum miRNA-629 levels correlated positively with poor OS and DFS [50].

Wang et al. showed that both miRNA-192 and miRNA-194 serum levels were lower in NSCLC patients than in healthy patients, both correlating with TNM stage, distant metastases, and pathological stages [51]. Exosomal levels were decreased in chemotherapy-resistant NSCLC patients compared with chemotherapy-sensitive NSCLC. In addition, serum miR-433 levels correlated positively with large tumor size, distant metastasis, advanced TNM stage, and poor prognosis [52]. Serum miRNA-590-5p levels were decreased in NSCLC patients compared with healthy controls. In addition, patients characterized by advanced stages III and IV had increased serum levels of this miRNA compared with NSCLC patients in non-advanced stages. A negative correlation of miR-590-5p has also been reported in the prognosis of NSCLC patients [53]. Serum miRNA-320a levels were decreased in NSCLC patients compared with healthy subjects, and were correlated with tumor size, TNM stage, and lymph node metastasis. In another study, the authors highlighted the problem of bone metastasis in NSCLC patients [54]. A cluster of serum miRNAs characterized by decreased expression of exosomal miRNA-574-5p and increased expression of both miRNA-328-3p and miRNA-423-3p was found in NSCLC patients with bone metastasis compared with NSCLC patients without bone metastasis [55].

### 4.2. Tissue miRNAs

Although studies are available regarding the role of several miRNAs in lung cancer tissue in the prognosis of patients with NSCLC, these have mainly evaluated cell lines in in vitro models, and few data are currently available on human lung cancer tissue [15].

One study found that miRNA-25 expression was increased in NSCLC lung tissue of patients compared with healthy subjects. In addition, miRNA-25 expression correlated positively with the radio resistance of the patients through the inhibition of B-cell antiproliferation factor 2 (BTG2) [56]. Another study found that miRNA-25 expression was increased in NSCLC lung tissue and the expression of this miRNA correlated negatively with OS of the patients and correlated positively with cell migration and invasion capacity through the activation of the large tumor suppressor homology 2/yes1-associated transcriptional regulator (LATS2/YAP) signaling pathway. In addition, in vitro use of a specific inhibitor of this miRNA reversed the effects seen in human lung tissue [57].

In vitro, the expression of miRNA-25-3p in A549 and H1299 cells correlated positively with cisplatin resistant and increased cell invasion of proliferation. These effects were mediated by the activation of miRNA-25-3p-induced by the PTEN/PI3K/AKT axis. Use of a specific inhibitor of this miRNA in the cell lines reversed the effects observed [58].

Similarly, in vitro increased expression of miRNA-15b in PC9-R and A549-R cells correlated positively with cisplatin resistance. In addition, PC9-R and A549-R cells miRNA-15b^−/−^ showed decreased cisplatin resistance [59].

Another in vitro study showed that miRNA-494 also correlated positively with proliferation, colony formation capability, and increased resistance to cisplatin-induced apoptosis in A549 and H460 cell lines. These effects were mediated by the inhibition of caspase-2 (CASP2) mediated by miRNA-494 [60].

The increased expression of miRNA-556-5p in tissue sampled from NSCLC patients correlated positively to cisplatin resistance, which was mediated by the inhibition of miRNA-556-5p-induced by the NLR pyrin domain-containing 3 family (NLRP3). In vitro, cells lines characterized by miRNA-556-5p^−/−^ showed reversed results with increased activation of NLRP3 and decreased cisplatin resistance [61].

Decreased expression of miRNA-30a in tissue sampled from NSCLC patients correlated with poor outcomes in term of lymph node metastasis, advanced TNM stage, and poor 5-year survival [62].

Decreased expression of miRNA-638 in lung cancer tissue obtained from NSCLC patients correlated with poor outcomes in term of increased tumor size and metastasis compared with lung samples expressing increased levels of miRNA-638. In vitro data confirmed the role of this miRNA, with lung cancer cell lines (A549, H1299, SPCA1, H1650, and H358) expressing decreased levels of miRNA-638 being characterized by increased proliferation and migration. Administration of the mimics of miRNA-638 reversed these effects [63].

## 5. Lung Microbiota in the Pathogenesis of NSCLC

The microbiota is a group of several microorganisms located in human organisms such as the gut and lungs and is represented mainly by bacteria but also fungi and viruses [64]. These microorganisms are part of the innate immune system that modulates the host immune response [65]. Dysbiosis is the condition characterized by an alteration in the microbiota in terms of composition, presence of different species of bacteria, or impaired production of metabolites from the microbiota. These conditions are correlated with an increased risk of several diseases but mainly chronic inflammatory disease [66]. In physiological conditions, there is a balance between microbial immigration and elimination in the lung, which modulates the composition of the microbiota [67]. In contrast, during pathological conditions in the lung, various growth conditions can alter the replication rates and the normal composition of microbiota [67].

Many studies have investigated the role of the microbiome in the gut in terms of the modulation of the innate immune response. However, fewer data are available on the lung microbiota, although evidence suggests that the lung microbiota also acts similarly in the modulation of the innate immune response [66]. Several studies have thus shown that lung microbiota plays a role in several lung diseases including lung cancer and is key to the initiation and progression of the disease. For example, in vivo data show that germ-free or antibiotic-treated mice models have a decreased risk of developing lung cancer, and this seems to be correlated to the decreased rate in Kras mutation and also p53 loss [68]. On the other hand, the lung microbiota can mediate an inflammatory response through the recruitment and activation of T cells, which decrease the development of lung adenocarcinoma [68].

Several mechanisms have been hypothesized to explain the increased risk of oncogenesis induced by lung dysbiosis, e.g., (a) impaired immune activation/tolerance, (b) chronic inflammation status, and (c) direct activation of several pro-oncogenic pathways [69]. Impaired immune activation/tolerance is also the most studied pathway and suggests how lung dysbiosis can impair the physiological stimulation of the immune system in the lungs. The absence or alteration of the microbial diversity can alter the antigen-presenting cell mechanism, decreasing the capacity of the immune cells to recognize the presence of neoplastic cells, which also suggests the important role of the lung microbiome in response to treatment with ICIs [70]. Conversely, increased bacterial growth mediates an enhanced stimulation of the immune cells with a higher rate of recruitment and activation of CD4 helper T cells releasing IL-17 (which has a key role in the pathogenesis of lung neoplasm [71]), through the activation of the Toll-like receptors (TLRs) pathway, involving the activation of nuclear factor-kappaB (NF-κB) and STAT3 [72].

Lung microbiota dysbiosis can induce chronic inflammation by DNA-chain damage and through the release of metabolites and genotoxins from affected commensal organisms. In addition, the immune/inflammatory cells stimulated by chronic inflammation mediated by dysbiosis can also produce and release reactive oxygen (ROS) and nitrogen species (RNS), which can mediate carcinogenesis through DNA damage and inhibition of the DNA-repair mechanisms [73,74]. On the other hand, oxidative stress can modulate several intracellular signaling pathways such as PI3K, ERK1-2, mTOR, and NF-κB, similarly to miRNAs and microbiota [74]. These intracellular signaling pathways not only have a key role in the pathogenesis of lung cancer but also in modulating the recruitment and activation of immune cells, and thus correlating to an impaired immune system that characterizes several lung diseases, including LC and COPD.

Several studies have evaluated the sputum derived from LC patients showing increased levels of several bacteria, above all *Streptococcus viridans* but also *Granulicatella adiacens*, *Streptococcus intermedius*, *Escherichia coli*, and *Acinetobacter junii* [75,76]. *Granulicatella*, *Abiotrophia*, and *Streptococcus genera* are increasingly being found in lung cancer patients [76]. One study showed that, when NSCLC cells were cultured in vitro with Gram-negative bacteria, especially *Escherichia Coli*, increased cell adhesion and migration capability were shown and these effects were mediated by the increased activation of TLR4 and the MAPK/ERK1-2 signaling pathways [77].

## 6. Interactions between miRNAs and Lung/Gut Microbiota in the Pathogenesis and Prognosis of NSCLC

Several in vivo studies (Table 1) have investigated the interaction between lung/gut microbiota and miRNAs in the development of NSCLC [10]. These studies have shown that the administration of several microorganisms such as *Bifidobacterium bifidum*, *Lactobacillus fermentum*, and *Lactobacillus salivarius* mediated the effects of tumor suppression through KRAS, STAT3, and ROCK1. These signaling pathways involved in lung cancer development were also modulated by several miRNAs, such as miRNA-143, miRNA-148a, and miRNA-223.In particular, miRNA-148a and miRNA-223 were found to increase and decrease, respectively, when *Bifidobacterium bifidum* was administered [10,78,79]. In addition, *Lactobacillus acidophilus* also induced lower levels of expression of the pro-oncogenic miRNA-9, thus decreasing the development of lung cancer [80].

Some evidence suggests how the dysbiosis in lung tissue involving several bacteria such as *Streptococcus*, *Prevotella*, and *Veillonella* may have a role in the development of lung cancer through the signaling pathways involving PI3K, MAPK, and ERK [64,81,82,83]. These pathways are modulated by several miRNAs, such as miRNA-126, miRNA-133b, and miRNA-145. However, no studies have yet verified whether these miRNAs are modulated by these microorganisms in the lung or gut. Only one pre-clinical study has suggested that miRNA-133 can regulate dysbiosis through the modulation of the TLR pathways [84].

Several studies have reported the interaction between microbiome and miRNAs in other cancers and also in other chronic lung diseases. These interactions act on the same intracellular signaling pathways, suggesting that they also play a role in the pathogenesis and evolution/prognosis of NSCLC [10,91].

For example, some in vivo studies have found that, by meditating the activation of the TLR4 signaling pathway, *Fusobacterium nucleatum* enhanced cell growth capacity in colorectal cancer in mice through the increased expression of miRNA-21 [36,85]. These data were also confirmed in humans showing that patients with colorectal cancer with high levels of miRNA-21 induced by *Fusobacterium nucleatum* were characterized by poor prognosis and greater chemoresistance [86].

Another potential interaction was shown in a pre-clinical model [in vitro (splenocyte cell lines) and in vivo (mice model)] of COPD, where *Lactobacillus acidophilus* lowered the expression of miRNA-21 and caused anti-inflammatory effects [92]. However, there were contrasting data regarding the interaction between *Lactobacillus acidophilus* and miRNA-21, showing that in an in vitro COPD model (human endothelial cell lines treated with LPS), *Lactobacillus acidophilus* increased the expression of miRNA-21, leading to an anti-apoptosis effect [87], thus suggesting that this axis may also have a role in cancers and in particular in NSCLC.

The expression of miRNA-375 in an in vivo colorectal mice model was modulated and decreased by microbiota dysbiosis, showing an enhanced proliferation of intestinal epithelial stem cell [93].

An in vitro study analyzing peripheral blood mononuclear cell (PBMCs) sampled from patients with systemic lupus erythematosus reported the decreased expression of miRNA-146a (which has a role in the regulation of several intracellular signaling pathways involved in lung cancer) compared with the PBMCs of healthy subjects. However, when these cells were cultured with *Lactobacillus rhamnosus* or *Lactobacillus delbrueckii*, the levels of miRNA-146a were restored by mediating anti-inflammatory effects [88]. These correlations between microbiota and NSCLC pathogenesis are summarized in Figure 1 and Figure 2.

## 7. Role of ICIs in the Treatment of NSCLC

ICIs are monoclonal antibodies that can target and inhibit the immune checkpoints, thereby leading to an enhanced activation and response of immune cells against tumor cells, particularly through the activation of cytotoxic T-lymphocytes [94]. Programmed death 1 (PD-1), programmed death-ligand 1 (PD-L1), and cytotoxic T lymphocyte-associated protein 4 (CTLA-4) inhibitors are the most commonly used ICIs in NSCLC, especially in the advanced stage [64,95]. These agents can reverse the immunosuppression characterizing the tumor microenvironment through the inhibition of T cell suppressing receptors when their corresponding ligands are bound to cancer cells. These monoclonal antibodies can also regulate the immune system response by increasing the recruitment and activation of effector T cells, especially CD8+ cytotoxic T cells, thus enhancing the identification of tumor-specific antigens and mediating tumor cell death [94].

## 8. Role of miRNAs as Predictive Factors for ICI Treatment in Patients with NSCLC

### 8.1. Circulating miRNAs

miRNAs expressed by tumor cells can modulate the composition of the tumor microenvironment [96], playing a key role in the interaction between tumor cells and immune cells [38,97].

Several studies have evaluated the levels/expression of circulating miRNAs as potential non-invasive predictive biomarkers for ICI treatment in NSCLC patients. However, these studies are limited by the poor standardization of the isolation methods for the miRNAs and the analysis of only small groups of patients.

In vitro, miRNA-34a mediates a decreased PD-L1 expression through p53 activation of A549, H460, and H1299 cell lines [98]. On the other hand, through the activation of the miRNA-200/ZEB1 axis, ex vivo miRNA-200b and miRNA-200c negatively modulate the onset of epithelial-to-mesenchymal transition (EMT) in NSCLC human tissue samples [99]. In vivo, miRNA-146a [100] and miRNA-155 [101] help recruit and activate lymphocytes Tregs, conversely miRNA-223 mediates both proliferation and differentiation of myeloid cells [102].

Several studies have thus analyzed the role of circulating miRNAs in patients with NSCLC to evaluate whether these miRNAs could be considered as potential predictive factors for the response to ICI therapy [4,5,6,7,8].

One study found that seven miRNAs, including miRNA-215-5p, miRNA-411-3p, miRNA-493-5p, miRNA-494-3p, miRNA-495-3p, miRNA-548j-5p, and miRNA-93-3p, in the serum of 51 NSCLC patients treated with nivolumab correlated positively with enhanced OS [6]. In another study, a “miRNA signature classifier” (MSC) (representing a large group of miRNAs analyzed specifically in this study) was identified by examining the expression of several miRNAs, including miRNA-16-5p, miRNA-451a, miRNA-486-5p and miRNA-92a-3p, miRNA-126-5p, miRNA-15b-5p, miRNA-221-3p, and miRNA-30b-5p, in combination with PD-L1 expression in 140 advanced NSCLC patients treated with ICIs. This study showed that intermediate- or low-risk levels in MSC and/or PD-L1 expression 50% were found in patients with high objective response rate (ORR) (*p* = 0.0024), progression free survival (PFS), and OS (*p* < 0.0001), whereas high risk levels in MSC correlated with poor prognosis [4].

Another panel of miRNAs in patients with NSCLC composed of miRNA-93, miRNA-138-5p, miRNA-200, miRNA-27a, miRNA-424, miRNA-34a, miRNA-28, miRNA-106b, miRNA-193a-3p, and miRNA-181 correlated with an increased response to nivolumab treatment [5]. Conversely, another study reported contrasting data with increased serum miRNA-200 levels correlating with decreased OS [7].

One study showed that decreased serum levels of miRNA-320b and miRNA-375 in advanced NSCLC patients treated with nivolumab were correlated with a significant clinical benefit such as complete or partial response. These findings can be explained by miRNA-320b activity mediating the activation of several proliferation genes, such as MYC or tubulin beta-1 (TUBB1), while miRNA-375 mediated the activation of immune-related genes such as JAK2, TGF-b2, the Wnt/b-catenine (FZD4, FZD8), and the Hippo pathway (YAP1), which are known to induce ICI resistance [103].

In addition, in advanced NSCLC patients, the increased serum levels of hsa-miR-320d, hsa-miRNA-320c, and hsa-miRNA-320b at baseline ICI treatment were correlated with poor response and progression disease. Patients characterized by a good response to ICI treatment also showed decreased serum levels of hsa-miRNA-125b-5p [104].

### 8.2. Tissue miRNAs

Very few studies have been carried out regarding the role of miRNAs in the prediction of ICI treatment in NSCLC; moreover, such data are derived from pre-clinical in vitro studies. For example, in vitro use of the lentivirus-transduced miRNA-3127-5p led to an increased expression of PD-L1 through an increased activation/phosphorylation of STAT3. These results were reversed in miRNA-3127-5p^−/−^ cell lines [105]. Another in vitro study showed that miRNA-505-3p can enhance CD8+Tcell activation and function, leading to an increased secretion of cytokines and enhanced cytotoxicity, thus suggesting that this miRNA may promote the effectiveness of ICI treatment in NSCLC [106].

## 9. Role of Lung Microbiota as Predictive Factor for ICI Treatment in Patients with NSCLC

Several studies have evaluated the role of microbiota in the prediction of cancer immunotherapy response. The microbiota enhances the effectiveness of immunotherapy in lung cancer patients. Currently, the most important data regarding the microbiota and ICI treatment in NSCLC regard gut microbiota. Gut dysbiosis (also when induced by antibiotics) can decrease the effectiveness of immunotherapy [64,107].

The varying effectiveness of ICIs in lung cancer patients may be correlated with the different composition of microbiota that affects the host immune landscape [108,109]. The gut–lung axis also has an important role and can modulate the effectiveness of ICIs in lung cancer patients. Patients with advanced NSCLC characterized by increased diversity of gut microbiota, when treated with anti-PD-1 agents showed an enhanced response compared with patients characterized by decreased diversity in gut microbiota. These patients with advanced NSCLC were also characterized by increased blood NK cell levels and memory T cells [68]. Another study showed that decreased levels of *Akkermansia muciniphila* were found in samples of NSCLC in non-responsive patients. In addition, in vivo use of fecal microbiota transplantation from patients responsive to immunotherapy led to the increased effectiveness of anti-PD-1/PD-L1 treatment in antibiotic-treated or germ-free mice models [89].

Several studies have confirmed that the diversity of gut microbiota (e.g., *Akkermansia* sp., *Bifidobacterium* sp., *Faecalibacterium* sp., *Enterococcus faecium*, and *Collinsella aerofaciens*) is positively correlated with immunotherapy responses in patients with advanced NSCLC. In contrast, antibiotics may decrease the effectiveness of the treatment as they encourage the development of gut microbiota dysbiosis [110,111,112,113]. Other data have shown that the use of *Clostridium butyricum* in patients with NSCLC before starting PD-1/PD-L1 target treatment significantly increased PFS and OS compared with NSCLC patients not treated with *Clostridium butyricum* [90].

## 10. Interactions between miRNAs and Lung Microbiota in the Response of ICI Treatment in NSCLC Patients

Currently, there are no data specifically regarding the miRNA/microbiota interaction in the modulation of ICI treatment in NSCLC (Table 1). However, some data regarding other cancers, such as colorectal cancer, have found a correlation between gut microbiota/dysbiosis and the expression of several miRNAs, which helps the ICI treatment response in NSCLC patients [114].

For example, an in vitro study showed that *L. mesenteroides* mediated the apoptosis of colon cancer cell lines (HT-29 cell lines) through the downregulation of miRNA-200b [115], which regulates PD-L1 expression and EMT capacity in NSCLC patients [98,99].

In contrast with the data shown in NSCLC patients, data regarding colorectal cancer have shown that a greater presence of *Akkermansia*, *Turicibacteracea* and *Coprococcus* (induced by a diet rich of Omega-3 and Omega-6) was correlated with the downregulation of miRNA-93 and miRNA-27 [116], which facilitates the effectiveness of ICI treatment in NSCLC patients.

In addition, data regarding the miRNA/microbiota interaction in intestinal bowel diseases (IBD) suggest another potential predictive interaction for ICI treatment in NSCLC. The use of *L. fermentum*, *L. salivarius*, and *S. boulardii* in a mice IBD model mediated decreased gut dysbiosis and downregulation of both miRNA-155 and miRNA-223 [79,117], which, as previously reported, play an important role in the modulation of immune response in NSCLC patients. Figure 1 and Figure 2 summarize the known and potential interactions between miRNAs and microbiota modulating both prognosis and ICI treatment response in NSCLC patients.

## 11. Conclusions

Currently, although knowledge on the pathogenesis of lung cancer has increased, and the use of new therapies such as ICIs has increased the survival rate of patients with lung cancer, lung cancer still has a high mortality and poor prognosis.

Several studies have evaluated the role of various biomarkers to predict both the prognosis and the effectiveness of lung cancer therapy. Interesting data come from studies analyzing the role of miRNAs and also the role of the gut/lung microbiota and the related microbiome. There is much evidence that both miRNAs and the gut/lung microbiome can modulate several key signaling pathways involved in the pathogenesis of lung cancer and ICI treatment response (which unfortunately cannot be used in all patients with lung cancer). These data suggest that both miRNAs and the microbiome represent useful biomarkers and also new potential targets for personalized therapy for lung cancer.

Some data also show that several miRNAs can modify the microbiome, inducing dysbiosis and vice versa, as these can target the same intracellular signaling pathways, thus suggesting a key role of the miRNA/microbiome axis as a new target in lung cancer therapy. However, data are still poor, especially regarding the role of the miRNA/microbiome axis in predicting the effectiveness of ICI treatment in NSCLC patients. Further studies are thus essential to clarify the importance of this axis, which could soon be used in clinical practice.

## Figures and Tables

**Figure 1 ijms-25-06685-f001:**
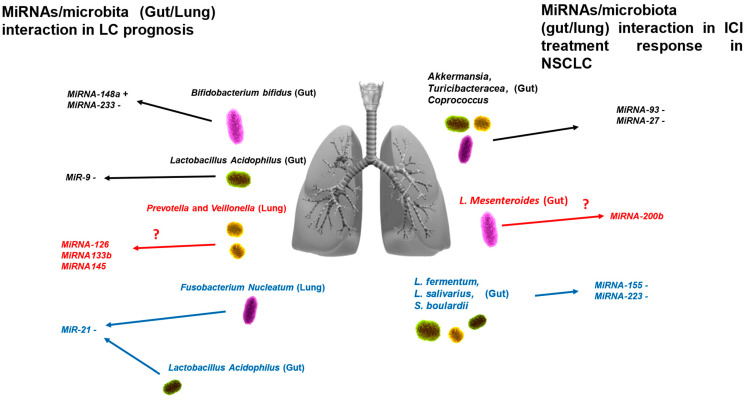
**Known and potential interactions between miRNAs and microbiota modulating both prognosis and ICI treatment response in NSCLC.** MiRNA/microbiota interactions modulating the prognosis in patients with NSCLC are shown on the left. miRNA/microbiota interactions modulating ICI treatment response in NSCLC are shown on the right. The interactions marked in black are already well known. The interactions marked in red are potential interactions that have been found in other cancers and that can modulate the same signaling pathways involved in NSCLC. The interactions marked in blue are well-known interactions that have been found in other chronic inflammatory diseases. Legend: +—increased expression, -—decreased expression; ?—unknown.

**Figure 2 ijms-25-06685-f002:**
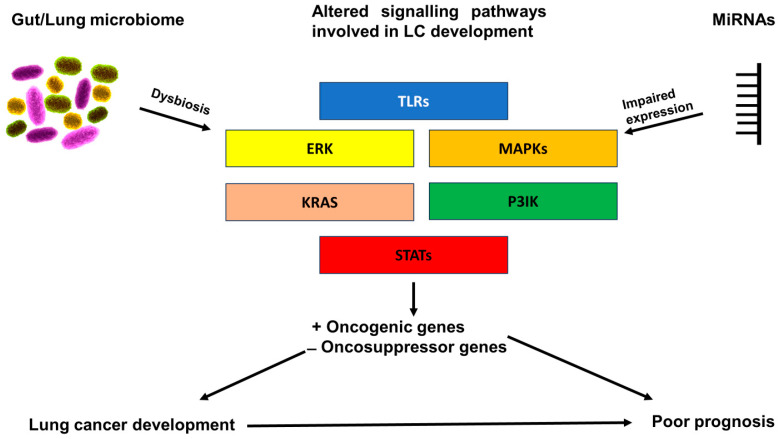
**Modulation of key intracellular signaling pathways involved in lung cancer development mediated by the interaction between microbiota and miRNAs.** There is an interaction between gut/lung microbiota and miRNA expression. Microbiota dysbiosis can alter the expression of several miRNAs, and vice versa. Both microbiota and the related microbiome and miRNAs play a crucial role in the modulation of key intracellular signaling pathways involved in lung cancer development. Some of these interactions may be correlated to poor patient prognosis with fewer benefits from the treatment and consequently lower survival rates.

**Table 1 ijms-25-06685-t001:** Interaction between miRNAs and microbiota in NSCLC prognosis and ICI treatment response.

Microbiota	miRNA +/−	Target Signaling Pathways	Biological Effects
*Bifidobacterium bifidum*, *Lactobacillus fermentum*, and *Lactobacillus salivarius* (gut)	miRNA-143+ miRNA-148a+ miRNA-223−	KRAS, STAT3, and ROCK1	Tumor suppression, enhanced cell differentiation, decreased cytoskeleton remodeling, and EMT process
*Streptococcus*, *Prevotella*, and *Veillonella* (lung)	miRNA126− miRNA133b+ miRNA145+	TLRs, PI3K, MAPK, and ERK	Decreased cell proliferation, increased OS
*Fusobacterium nucleatum* (lung)	miRNA21+	TLR4	Poor prognosis and chemoresistance
*Lactobacillus Acidophilus* (gut)	miRNA21+/−	TLRs	Anti-inflammatory effects (−), anti-apoptotic effect (+)
*Lactobacillus rhamnosus* or *Lactobacillus delbrueckii* (gut)	miRNA-146a+	?	Anti-inflammatory effects
*L. mesenteroides* (gut)	miRNA200b−	PD-L1	Cancer cell apoptosis and decreased EMT capacity
*Akkermansia*, *Turicibacteracea* and *Coprococcus* (gut)	miRNA-93− miRNA-27−	PD-L1	Decreased ICI treatment response

Data obtained from [10,31,64,68,78,79,80,81,82,83,84,85,86,87,88,89,90]. Legends: +—increased expression, −—decreased expression; +/−—contrasting data in the literature.

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
