# Peer review of "MiRNAs and Microbiota in Non-Small Cell Lung Cancer (NSCLC): Implications in Pathogenesis and Potential Role in Predicting Response to ICI Treatment"

_ijms, 2024, doi:10.3390/ijms25126685_

Round 1
Reviewer 1 Report
Comments and Suggestions for Authors
In this study, the authors read a large number of literatures, and through systematic analysis and summarization, they elaborated the impact of MiRNAs and microbiota in the pathogenesis of non-small cell lung cancer (NSCLC) and predicted their potential roles in the therapeutic response to ICIs. From this, it is concluded that miRNAs and lung/gut microbiomes can play an important role in predicting prognosis and treatment effectiveness of ICIs in NSCLC, and that these axes may be novel targets for the treatment of NSCLC. Overall, the authors provide a good summary of the relevant studies and point the way for future research in NSCLC treatment and prognosis.
The following are some comments and suggestions that are given to improve the manuscript:
1. The format of the references in this article is not uniform, for example, the authors are not listed in all of the references in line 579, and they should be listed in all of the references. The reference format should be modified as “Boeri M, Milione M, Proto C, Signorelli D, Lo Russo G, Galeone C, Verri C, Mensah M, Centonze G, Martinetti A, Sottotetti E, Pastorino U, Garassino MC, Sozzi U, Sottotetti A, Sottotetti E. Garassino MC, Sozzi G. Circulating miRNAs and PD-L1 Tumor Expression Are Associated with Survival in Advanced NSCLC Patients Treated with 580 Immunotherapy: a Prospective Study. Clin Cancer Res 2019, 25, 2166-2173, doi:10.1158/1078-0432.CCR-18-1981”. There are many other similar issues in the text, please check and revise in full.
2. Subheadings “3.1”, “3.2” and “3.3” in the text are inconsistent with the content of the subsequent expressions, and it is suggested that they be revised. 3.1 MiRNAs involved in tumor-suppressive signalling pathway” should be changed to ‘MiRNAs are involved in the regulation of tumor-suppressive signalling pathway’. tumor-suppressive signalling pathway”. “3.2 MiRNAs involved in oncogenic signaling pathways” is suggested to be changed to “Involvement of miRNAs in the regulation of oncogenic signaling pathways", ”3.3 3.3 MiRNAs involved in signalling pathways with dual effects on tumorigenesis “ is suggested to be changed to ‘MiRNAs have dual roles in tumorigenic signaling pathways’.
3. There are many other expressions in the article that are not relevant enough, and the authors are advised to read the whole article carefully and check themselves.
Comments on the Quality of English LanguageMinor editing of English language required
Author Response
We thank the referee for his/her comments, we have changed the text in following the comments, the changed are marked in green:
1- Thank you for your comment. We have revised the references list. The references are correclty inserted following the journal style. We have used the software "Endnote" to avoid errors in the syle.
2- Thank you for your comment. We have changed the subparagraph titles as suggested.
3- Thank you for you comment. We have totally revised the text
Reviewer 2 Report
Comments and Suggestions for Authors
Review report on ijms-3044813 manuscript entitled on ‘MiRNAs and Microbiota in NSCLC: a focus on their implications in pathogenesis and on their potential role in predicting response to ICI treatment’
The authors seem to have sought to explore the impact of microRNAs, microbiomes, and microbiomes related microRNAs as the pathogenesis of NSCL, and the possibility of prognostic factors in Non-small cell lung cancer (NSCLC), and the possibility of the predictive factors of NSCLC treated with immune checkpoint inhibitors (ICIs) using microRNAs, microbiomes, and microbiomes related microRNAs.
MicroRNAs and microbiomes are emerging research topics in current cancer researches, and in particular, in research on microbiomes and cancer, many studies have been reported focusing on GI tract cancers including colon cancer, but there have been few studies on the lung microbiome of lung cancer. Therefore, if this study could have precisely and accurately reviewed the lung microbiomes of NSCLC and its related microRNA, it would have been a pioneering study with great significance.
However, the gut microbiome and lung microbiome are described mixed (Figures 1 and 2), which may cause confusion and ambiguity for readers by referring to the gut and lung microbiomes together. In addition, this study cites many previous studies on microRNAs, but lacks a clear explanation of the cell lines, tissues, and species of the microRNAs, and does not accurately describe whether the study is in-vitro or in-vivo, or whether the microRNA study is human or mouse. The fact that references are mentioned all at once in Table 1 is an example of how vague and insufficiently detailed the study is.
Although microRNA and microbiome are currently emerging research topics, they are also fields in which much has not yet been discovered. Therefore, in order to assert the authors' opinions through literature reviews, it is necessary to clearly mention the parts that have been clearly and precisely identified.
It is suggested that narrowing the research topic more clearly and focusing on microRNAs related to the lung microbiomes will reveal more significance of the study. It is necessary to narrow down the research topic and clearly describe the research results.
Comments on the Quality of English LanguageIt seems necessary to check for misspelled words and indicate the authors' affiliations in English.
Author Response
We thank the referee for his/her comments, we have changed the text in following the comments, the changed are marked in green:
General comments: Thank you for your comment. We have specified in both figures and table if the microbiota are derived from gut and lung. In addition, we have emphatized in the text if the cited studies analyzed human tissues or were pre-clinical studies (in vivo o in vitro). In particular, we have specifed the animal model for the in vivo studies and the cell lines used for the in vitro studies.
Finally, we have already focused on the well-known microbiota/miRNA in lung cancer. However, we have evaluated the microbiota/miRNAs interaction seen in other cancers or in other lung diseases as we strongly think that these interaction may stimulate novel studies evaluating these interaction in lung cancer to found novel useful prognostic and predictive biomarkers.
In additionn, we have exstensively revised the manuscript to improve the English language.
Reviewer 3 Report
Comments and Suggestions for Authors
I thank the Editorial Board of the journal IJMS for the opportunity to review the manuscript entitled ‘’MiRNAs and Microbiota in Non-small cell lung cancer (NSCLC): a focus on their implications in pathogenesis and on their potential role in predicting response to ICI treatment’’ by Francesco Nucera et al.
The authors address a topic of considerable interest, both purely biological and of great clinical impact.
General comment/question of interest: although a rather infrequent occurrence, some lung transplant patients develop lung cancer after transplantation. Are there any data in the literature on this? Are there known interactions between specific miRNAs and microorganisms in the gut microbiota? Section 4.2: The authors differentiate between circulating and tissue miRNAs. The second category includes all non-circulating miRNAs, how are cellular miRNAs classified? When I think of circulating miRNAs, I instinctively think of circulating molecules, similar to free DNA fragments. May I ask the authors for clarification on this?
Last general comment on paragraph 12: From the title ‘Highlights’ I expect a short and quick overview of the main concepts, instead the authors repeat concepts already extensively discussed in the text. I suggest, should the authors retain this part, to make it more schematic and less discursive, otherwise it turns out to be just a repetition of previously written concepts.
Here are some comments and suggestions to improve the readability and comprehension of the manuscript.
Lines 228-230: Is anything known in the literature about immunotherapy or chemo-immunotherapy?
Lines 195 and 243: The authors cite a couple of references to advanced stages. Is there any data in the literature on early stages (I and II)? If so, can they explain?
Line 266: The microRNA identifier (miR) should be written uniformly; specifically miR or MiR. I ask the authors to check in the text and standardise if necessary.
In Table 1: The second column (miRNA +/-) shows the expression of miRs in tissues? If not, please explain further.
Figure 1 and also in general: Few data have been produced for SCLC; have the authors evaluated the topic, even if only in terms of comparison, NSCLC vs SCLC?
Figure 1: Can the authors explain the red and blue colours of the arrows? The difference is not clear to me. Also, what does the ‘-’ sign in the figure represent/distinguish? I suggest explaining in the legend, otherwise it is difficult to understand the figure, which is also very supportive in the intricate understanding of the various interactions.
Lines 397-400: What do the authors mean by ‘interaction marked in red or in black’? If the colours refer to the previous figure, they must be written in the preceding legend, otherwise I ask for a clearer explanation.
Line 434: Can the authors better explain the concept of ‘miRNA signature classifier’? I think it represents the expression of different miRNAs, but it is not clear to me whether it concerns those specifically listed in lines 435-436 (NSCLC patients treated with ICIs) or refers to miRNAs more generally?
Paragraph 9, lines 468-472: The authors mention the terms microbiome and microbiota interchangeably. However, the two terms refer to biologically different concepts: the microbiota represents the set of microorganisms, the microbiome represents the genetic make-up of the microbiota. I suggest that the authors verify the correctness of the contexts in which they use the two terms.
Lines 556-559: Again, with regard to the lung, the authors mention several chronic diseases (e.g. asthma, COPD), but post-transplant organ rejection is also a cause of high mortality in patients. In these patients, in fact, the immune response is already severely compromised by the immunosuppressive therapies they are subjected to. I wonder if, in your opinion, any interactions between miRNAs/regulated pathways and the gut microbiome/microbiota could contribute to altering an already highly compromised immune response.
Comments on the Quality of English LanguageIn general, I recommend a thorough revision of the English language, in some places the text is not easy and straightforward to understand. Furthermore, there are many typos scattered throughout the text (typoes) that need to be corrected.
Author Response
We thank the referee for his/her comments, we have changed the text in following the comments, the changed are marked in green:
General comments:Thank you for your comments. We have no found data regarding lung cancer and lung transplantation. Anyway, this was not the topic of our manuscript. In addition, as already reported in our manuscript some interaction between microbione and lung cancer are known and these are reported in the revised figure. The tissue miRNA are molecules structurally similar with circulating miRNA, however, these are obtaind direclty through the RNA-sequancing from lung tissue compared the circulating miRNA obtainsed from serum. We have enhanced this concept in the manuscript. Finally, we have revised the highlight. Now are more schematic and focused on the topic.
1- Thank you for your comment. We have not found any data regarding the correlation between miRNA-629 and chemo/immunotherapy in cancers in general.
2- Thank you for your comment. Regarding the ref 55, the topic of the study was specifically the presence of bone metastatis correlated with the expression of specific miRNAs. Thus, the involvment of I-II stage NSCLC patients could represent an important bias as the absence of bone metastatis in these patients could be correlaed to the early stage, but not with the expression of these miRNAs. Regarding ref 41, the study have evaluated both NSCLC patients with stage I-II, but also patients with stage III-IV showing that the expression of specific miRNAs was increased in late stage NSCLC patients. Thus, also early stage NSCLC patients were evaluated. In addition, in refs 39,40,42,43,45,46,47,48,49.50,51,53,54 different stage (including I-II) of NSCLC patients were evaluated. We have resived this part of the manuscript to emhatize the differences of several miRNAs between early and late stage NSCLC.
3- Thank you for your comment. We have corrected the acronymous.
Table 1: Thank you for your comment. We have adda Lenged in the table to explain this poinf as follow: "Legend: +:increased expression, -: decreased expression; +/-: contrasting data in literature."
Figure 1: Thank you for your comment. We have revised both the figure and the figure legend.
4- Thank you for your commets. These sentences were regarding the figure legend. As previously reported we have revised the figure legend.
5- Thank you for your comment. “miRNA signature classifier” (MSC) represented a large group of miRNAs analyzed specifically in the cited stuy, we have clarify this point in the text.
6- Thank you for your comment. We agree with the referee. We have revised the manuscript and we have correclty used the term micrbiota or microbiome.
7- Thank you for your comment. We agree with the referee. Indeed, in the manuscript is reported many times that the interaction between miRNAs/microbiota can modulate several intracellular signalling pathways that have a key role in pathogenesis of lung cancer, but also can modulate recruitment and activation of immune cells. We have focused this point in the text. However, as previoulsy reported, there are no data redarding lung translantation.
Finally, we have extensively revisec the text to improve the English language
Round 2
Reviewer 2 Report
Comments and Suggestions for Authors
none
Author Response
We thank the referee for his/her comments improving our manuscript
Reviewer 3 Report
Comments and Suggestions for Authors
I thank the authors for the thorough revision of the manuscript.
I only have one suggestion concerning the section entitled ‘Highlights’: in my opinion, it is still very scattered. I would suggest that the authors greatly reduce, if not even remove, lines 552 to 560; these are in fact, widely known concepts in the literature. I would rather shine the light on the data they have obtained. Consider this suggestion.
Author Response
We thank the referee for his/her comment.
We have modified the "Highlights" as suggested, we hope that now the manuscript is suitable for the pubblication